# Types, design, implementation, and evaluation of nutrition interventions in older people in Africa: A scoping review protocol

Anthony Manyara[1]*, Tadios Manyanga[2], Rudo Chingono[2], Shane Naidoo[3], Kate Mattick[2,4], Grace Pearson[1], Opeyemi Babatunde[5], Niri Naidoo[6], Kate A. Ward[7,8], Celia L. Gregson[1,2]

1 Global Health and Ageing Research Unit, Bristol Medical School, University of Bristol, Bristol, United Kingdom, 2 The Health Research Unit Zimbabwe, Biomedical Research and Training Institute, Harare, Zimbabwe, 3 Health and Rehabilitation, University of Cape Town, Rondebosch, Cape Town, South Africa, 4 Brighton and Sussex Medical School, Brighton, United Kingdom, 5 School of Medicine, Primary Care Centre Versus Arthritis, Keele University, Keele, United Kingdom, 6 Division of Physiotherapy, Department of Health and Rehabilitation Sciences, University of Cape Town, Observatory, Cape Town, South Africa, 7 MRC Unit The Gambia, London School of Hygiene and Tropical Medicine, Banjul, The Gambia, 8 MRC Lifecourse Epidemiology Centre, Human Development and Health, University of Southampton, Southampton, United Kingdom

* Anthony.Manyara@bristol.ac.uk

**Data Availability Statement:** All data are in the paper and its Supporting Information files.

## Abstract

### Introduction

Africa's older population is increasing and this, necessitates the development of interventions to promote healthy ageing. Nutrition is a key determinant of healthy ageing and local contextual evidence is needed to inform nutritional intervention development in Africa. There are already reviews on nutritional status and food insecurity in older adults in Africa. However, a synthesis of nutrition interventions targeting older people specifically, is lacking. Therefore, this protocol describes a scoping review that aims to systematically synthesise current evidence on nutrition interventions for older people in Africa.

### Methods

The review will involve: a review of available reviews on nutrition in older people in Africa (Phase 1); a review of nutrition interventions developed or implemented among older people in Africa (Phase 2); and consultation with local nutrition stakeholders in Zimbabwe, and The Gambia to contextualise Phase 1 and 2 findings and solicit insights not in the published literature (Phase 3). Searches for Phases 1 and 2 will include bibliographic databases (MEDLINE, EMBASE, Web of Science, African Journals Online, African Index Medicus) and grey literature sources (i.e. relevant websites). Title, abstract, and full-text screening will be conducted in duplicate, data extracted using piloted tools and findings summarised using descriptive statistics and narrative text. Phase 3 will be conducted using hybrid workshops, audio-recorded, detailed notes taken, and findings combined with those from Phases 1 and 2.

**Funding:** This work is supported by the National Institute for Health Research (NIHR) (using the UK's Official Development Assistance (ODA) Funding); CG, AMM, TM, RC, GP are funded via NIHR302394. The views expressed are those of the authors and not necessarily those of the NIHR or the Department of Health and Social Care. For the purpose of Open Access, the author has applied a CC-BY public copyright licence to any Author Accepted Manuscript version arising from this submission. The funders will have no role in study design, data collection and analysis, decision to publish, or preparation of the manuscript.

**Competing interests:** The authors have declared that no competing interests exist.

## Conclusion

The findings of this review will summarise current evidence on nutrition in older people in Africa and inform nutrition intervention development. The findings will be presented in conferences, meetings, and published open access.

## Trial registration

This scoping review has been registered in OSF, registration DOI: https://doi.org/10.17605/OSF.IO/FH74T.

## Introduction

The global population is increasing through an unprecedented and sustained demographic transition, characterized by increasing longevity and reduced fertility rates [1]. In 2017, the global population was 7.6 billion, and is projected to peak at 9.7 billion in 2064 before falling to 8.8 billion by 2100 [2]. Older adults comprise an ever growing proportion of the current global population [1, 3]. As such, multimorbidity and disability levels are expected to rise, creating pressure on health and social care systems, thus emphasising the vital role of preventive interventions [4] to maximise healthy ageing i.e., developing and maintaining physical and mental capacities for both healthy and unhealthy people, and across the life-course, to improve well-being in older age [3].

The number of Africans aged ≥60 years is expected to triple in the next three decades: from 74.4 million in 2020 to 235.1 million in 2050 [5]. This reality calls for interventions to facilitate healthy ageing and to cushion economies and health systems, from negative impacts of rapid and unhealthy ageing [6]. To ensure cost-effectiveness and reduce resource waste, interventions should be informed by local and contextual evidence [7]. However, research and evidence synthesis on health ageing in older adults in Africa are scarce, although emerging. For example, Naidoo et al. [8] are currently synthesising current evidence on physical activity interventions in older people in sub-Saharan Africa (SSA) to inform future interventions [8]. Apart from physical activity [9], diet is a fundamental determinant of, and risk to, healthy ageing [9]. Healthy diets (e.g., high in fruit and vegetables, fish, whole grains) are associated with longevity, and better cognitive, musculoskeletal and cardiometabolic health [10], while less healthy diets (e.g., high in sugar and sodium, low in fruit and vegetables) are important risk factors of chronic diseases such as cancer, cardiovascular disease, and type 2 diabetes [11].

Our preliminary literature search has found that available reviews on nutrition in older adults in Africa have synthesised descriptive studies and focussed on: nutrition and health status [12], nutrition status and associated factors [13] and food insecurity [14]. Of these three reviews, all urgently called for nutritional interventions: however, our preliminary literature search did not identify any existing or ongoing reviews focussed on any aspect of nutrition interventions targeted at older adults in Africa. Nevertheless, we identified a few studies documenting the implementation of nutrition interventions among older adults in Africa. Therefore, to inform current and future nutrition interventions in older adults in Africa, this protocol describes a scoping review aiming to synthesise available evidence on nutrition and nutrition interventions for older people. Particularly, the review will seek to explore the types, design, implementation, delivery, and evaluation of such interventions. The review will comprise of three interlinked phases described in more detail in the methods section: Phase 1

(review of reviews on nutrition in older people in Africa); Phase 2 (review of empirical studies describing nutrition interventions (both nutrition-specific (e.g. supplementation) and nutrition-sensitive (e.g. cash transfers)) in older people in Africa); and Phase 3 (consultation with local stakeholders on Phase 1 and 2 findings to gain contextual insights not in the published literature). The scoping review will inform the development of nutrition interventions in Africa, and more specifically and immediate a healthy ageing intervention in Zimbabwe as part of a wider project on health ageing in Africa.

## Methods

A scoping review is appropriate for evidence synthesis in this case, as this design is ideal for areas where evidence is emergent, few randomised trials relevant to a given context exist, and where the aim is to generate evidence beyond intervention effectiveness [15]. This scoping review will be conducted using steps from a framework proposed by Arksey and O'Malley [16] and further proposals by Levac et al. [15] as described below.

### Research questions

Table 1 shows the intervention characteristics, research questions, and related review phases.

### Literature search and selection

**Phase 1: Review of nutrition reviews.**   Published reviews will be searched in five databases MEDLINE and EMBASE (through Ovid); Web of Science, African Index Medicus and African

**Table 1. Intervention aspects and research questions to be addressed by the review and relevant review phases.**

| Intervention characteristic | Review questions (RQs) | Description | Phase involved in answering RQs |
|---|---|---|---|
| **Types** | 1. What types of nutrition interventions have been mplemented for older adults living in Africa? | Interventions which have been implemented in empirical studies will be categorised based on characteristics such as function (e.g., behaviour change, nutrient supplementation); disease or condition targeted; target prevention level (primary, secondary, tertiary) * | Phase 2, and 3 |
| | 2. What are the components of implemented interventions? | Components of interventions† e.g., leaflets and group counselling for education interventions | Phase 2 |
| **Design** | 3. Who and what should nutrition interventions target? | Based on reviews of descriptive studies, describe who (e.g., sociodemographic of older adults) and what (e.g., sarcopenia, malnutrition, obesity) the nutrition interventions target | Phase 1 |
| | 4. How have interventions been designed? | Explore evidence base, theory of change and/or behavioural framework, end-user engagement used to develop interventions. | Phase 2 |
| **Implementation** | 5. How have interventions been (or can be) implemented? | Implementation characteristics including setting (community, rehabilitation); intervention duration; approach (individual versus group) and mode of delivery (e.g., peers, health workers) | Phase 1, 2, 3 |
| | 6. What implementation approaches influence intervention uptake, acceptability, feasibility, and effectiveness? | Implementation approaches (e.g., incentivisation) and contextual factors that influence uptake, acceptability, feasibility, and effectiveness | Phase 2 |
| | 7. What is the cost and sustainability of implemented interventions? | Costing of interventions, or components of intervention, and/or their sustainability | |
| **Evaluation** | 8. How have the interventions been evaluated? | Study types (e.g., randomised controlled trials, quasi-experimental, process evaluation, cost evaluation) and primary and secondary outcomes used to evaluate interventions and their delivery. | Phase 2 |

*Primary prevention aims to prevent development of disease or clinical conditions by targeting risk factors; Secondary prevention aims to target pre-symptomatic disease and prevention of disease symptoms through interventions such as screening; Tertiary prevention aims to target developed disease and disability to promote functional ability, increase quality of life, limit disability and delay death [17].

†The review will not be interested in specific foods as this will vary considerably

Journals Online. Systematic searches, without time restrictions, will combine four concepts and their synonyms: older adults, nutrition, review, and Africa. An initial MEDLINE search strategy, informed by published and relevant reviews [18, 19], has been developed by one author (AMM) and checked by an experienced systematic reviewer and librarian–see S1 File. This search strategy will be adapted for EMBASE (through OVID), and the Web of Science; however, the African Journals Online and African Index Medicus databases do not allow for advanced searching, so simple searches using the website search function will be used. The searches will involve multiple searches combining relevant concepts using 'AND' (e.g., nutrition and review and older adults; diet and review and elderly). For feasibility, only the first 100 results of each search output (where multiple searches are used) will be screened [20, 21]. Furthermore, to capture reviews that have been conducted by non-academic authors (such as non-governmental organizations), relevant website searches (e.g. HelpAge International website) will be performed (using the simple search strategy already described) and the first 100 search outputs of each search screened [20, 21].

Search results from OVID and Web of Science will be exported to Endnote for removal of duplicates and then uploaded to Rayyan [22] for title, abstract, and full-text screening by two independent reviewers. Disagreements in screening decisions will be resolved through discussion and/or involving a third reviewer. For efficiency, searching and screening of records (using title and text underneath and if needed accessing full text) in African Journals Online, African Index Medicus and websites will be conducted simultaneously by one author. Records will be included if they document a review of any kind synthesising African evidence on nutrition in older adults.

**Phase 2: Review of nutrition interventions.** This Phase will involve searches of empirical studies describing implemented or planned nutrition interventions targeted at older adults in Africa. The literature search and selection approach will be similar to that described for Phase 1, albeit with a different search strategy. In addition to databases used for Phase 1, we will also seek protocols for ongoing research by searching trials registers including the International Clinical Trials Registry Platform and clinicaltrials.gov. S1 File shows the MEDLINE search strategy for this Phase which will combine four concepts and their synonyms: nutrition; intervention; older adults; and Africa. The search strategy was developed using synonyms used in published reviews [18, 19]. Like Phase 1, search results from MEDLINE, EMBASE, and Web of Science will be exported to Endnote for duplicate removal and uploaded to Rayyan [22] for title, abstract, and full-text independent screening by two reviewers. Similar to Phase 1, searches and screening in African Journals Online, African Index Medicus, and grey literature will be conducted simultaneously by one author. Records will be included if they describe any aspect of nutrition interventions (type, design, implementation, evaluation) piloted or implemented in older adults in Africa. Both published articles and grey literature with data on any research question will be included; however, reviewers will decide on whether to include certain grey literature types such as conference abstracts based on comprehensiveness and added value of data presented.

For both Phase 1 and 2, databases' and grey literature search inclusions will be complemented by screening reference lists of included full texts for relevant records. Additionally, using our professional contacts and networks, we will solicit relevant resources from organisations working on nutrition in older adults and healthy ageing interventions in Africa.

Table 2 shows additional inclusion information based on Population, Intervention, Comparison, Outcomes, and Study design criteria.

**Table 2. Inclusion criteria based on the population, intervention, comparison, outcomes, and study design criteria.**

| Criteria | Description |
|---|---|
| Population | Older adults who are community dwelling or residing in any long-term care facilities such as nursing homes, retirement centres. Inclusion of older adults in long-term facilities promotes equity and generalisation of findings [23]. We define older adult as those aged ≥50 years*. Studies will be included if the mean age or 70% of the study population is ≥50 years. The study has to be conducted, fully or in part, in an African country. |
| Intervention | Any nutritional intervention or multicomponent interventions with a nutrition component. Studies will be included if they describe any of the findings relevant to the research questions including intervention development process, piloting, feasibility testing and evaluation of interventions. |
| Comparison | Not relevant for this review |
| Outcomes | Not relevant for this review, see below for study types. |
| Study design | Any study design including randomised trials, quasi experiments, qualitative studies such as implementation studies, process evaluations, case studies with relevant data to on research questions. We will also include non-research reports that are relevant to the research questions. |

*An age of 50 years was used to define an older adult in Africa given that life-expectancy is lower in the region compared to high-income countries where an age of 60 or 65 years is traditionally used to define 'older'.

## Data extraction and charting

Two data types will be extracted from all records included in Phase 1 and 2: 1) record characteristics; and 2) data relevant to research questions. Record characteristics will include record type (journal publication, report, grey literature), publication year, country (for Phase 2), authors' names and affiliations (e.g., academic institution, non-governmental organisation), funding and potential conflicts of interest. Data relevant to research questions will be collated verbatim under each research question. A data extraction tool (in Microsoft Excel) will be piloted with a subset of the included records from each phase and modified if necessary. At the start of data extraction, a subset of records will be independently extracted by two authors and discussed for agreement, after which extraction will be completed by one author for each record and checked for accuracy by a second author.

## Synthesis of Phase 1 and 2 findings

All data will be extracted and analysed in Microsoft Excel. Record characteristics will be analysed by one reviewer using frequency counts and percentages and presented in text, tables or graphs. Data relevant to research questions will be analysed using a simple form of thematic analysis [24] and presented in frequency counts and percentages, text or appropriate visualisations. Two authors will independently read through collated text and identify relevant codes/ themes after which they will meet to discuss a coding framework that will be applied to all the data by one author. Table 3 summarises the analysis and synthesis that will be used for each of the review questions.

## Phase 3: Consultation

Consultation is an optional yet vital component of scoping reviews [15, 16]. The purpose of the consultation phase will be twofold: 1) knowledge transfer of findings from Phase 1 and 2; and 2) solicitation of local contextual perspectives based on presented findings and identification of important local aspects of nutrition interventions for older adults that are not in published evidence [15]. Consequently, the consultation will be conducted as a half day workshop and structured to include presentation of main findings from Phase 1 and 2 and then a

Table 3. Summary of analysis and synthesis for each of the review questions.

| Intervention characteristic | Review questions (RQs) | Analysis/synthesis |
|---|---|---|
| Types | | |
| | 1. What types of nutrition interventions have been implemented for older adults living in Africa? | Descriptions of different intervention types, diseases or outcomes targeted, and purpose of intervention *i.e.*, prevention or treatment |
| Design | 2. What are the components of implemented interventions? | Descriptions of components of interventions *e.g.*, for education interventions, written information, or oral group counselling |
| | 3. Who and what should nutrition interventions target? | **Who:** Descriptions of factors associated with nutrition and nutrition outcomes, categorised into four nutritional influences drawing from the framework by Story et al. [25] <br> **What:** pooled prevalences of nutrition outcomes (*e.g.*, malnutrition, obesity, weight loss) from meta-analyses |
| Implementation | 4. How have interventions been designed? | Descriptions of evidence base supporting intervention formative work, theory of change behavioural framework supporting intervention and end-user engagement during intervention (co-)development |
| | 5. How have interventions been (or can be) implemented? | Descriptions of implementation characteristics including setting (*e.g.*, community, hospital); intervention duration and dose; approach (e.g., individual versus group) and mode of delivery (e.g., peers, health care workers), complex or stand-alone intervention |
| | 6. What implementation approaches influence intervention uptake, acceptability, feasibility, and effectiveness? | Quantitative and qualitative data from process evaluations on implementation approaches (e.g., incentivisation) and contextual factors that influence uptake, acceptability, feasibility, fidelity and effectiveness |
| Evaluation | 7. What is the cost and sustainability of implemented interventions? | Costs of interventions, or components of intervention (e.g., direct health costs, patient out-of-pocket costs), and descriptions on their sustainability and health economic modelling of future intervention / scale-up. |
| | 8. How have the interventions been evaluated? | Description of study designs (e.g., randomised controlled trials, quasi-experimental, process evaluation, cost evaluation), primary and secondary outcomes used to evaluate interventions and their delivery, consideration of risks of bias. |

discussion. The consultation exercise will be conducted in Zimbabwe where an intervention for older adults (with a nutritional component) will be developed Eight to ten stakeholders will be identified through professional and local networks after stakeholder mapping and will focus on local nutrition experts and older adults, and/or community-based representatives, who will be potential intervention end-users. In-country project coordinators will invite the stakeholders for the consultation.

Consultations will be audio recorded (with stakeholders' consent) and will be used to make detailed notes during the discussions; insights gathered will be thematically analysed [24] and integrated with Phase 1 and 2 findings.

## Dissemination of findings

Review findings will be disseminated through conference and meetings' presentations; and an open-access publication reported using Preferred Reporting Items for Systematic reviews and Meta-Analysis for Scoping Reviews [26].

## Ethics

This is mainly a literature review-based study and ethical approval is not required. The consultation phase of the project will take the form of stakeholder workshops and will not collect personal data. This forms part of the formative phase of the **K**eeping **O**lder people healthy: de**S**igning and evaluating effective **HE**alth **S**ervices to maintain functional **AbI**lity (KOSHE-SAI) study, for which governance and ethical permissions have been approved by the

Biomedcial Research and Training Institute (BRTI) (ref: AP191/2023), City of Harare Health (ref:3/7), the Medical Research Council of Zimbabwe (MRCZ) (ref:A/3077) and the Research Council of Zimbabwe (Ref: MRCZ/A/3077).

## Conclusion

This review will summarise all previous reviews related to nutrition in older adults, and empirical studies describing development or implementation of nutrition interventions in older adults in Africa. Furthermore, local stakeholders will be consulted to enrich the findings from the published literature. Evidence from this review will inform the development of a healthy ageing intervention for use in Zimbabwe and, if it proves successful, for future nutrition interventions targeting older adults in Africa.

### Study status

Completion is anticipated by July 2024.

### Strengths and limitations

Search and inclusion of journal articles and grey literature will provide a comprehensive synthesis of evidence on nutrition in older adults in Africa. Additionally, consultation with local stakeholders will contextualize the findings and provide insights that are not in the published literature. However, consultation exercises will only be conducted in two countries which may not be generalisable to the African region.

### Supporting information

**S1 Checklist. PRISMA-P 2015 checklist.**
(DOCX)

**S1 File. Search strategies for Phase 1 and 2 in MEDLINE.**
(DOCX)

## Acknowledgments

The authors would want to thank systematic reviewers at the Musculoskeletal Research Unit, Bristol Medical School, Jane Dennis, and Andrew Beswick, for their help in the development of the search strategy.

## Author Contributions

**Conceptualization:** Anthony Manyara, Celia L. Gregson.

**Funding acquisition:** Celia L. Gregson.

**Investigation:** Anthony Manyara, Tadios Manyanga, Rudo Chingono, Shane Naidoo, Kate Mattick.

**Methodology:** Anthony Manyara, Tadios Manyanga, Rudo Chingono, Shane Naidoo, Kate Mattick, Grace Pearson, Opeyemi Babatunde, Niri Naidoo, Kate A. Ward, Celia L. Gregson.

**Writing – original draft:** Anthony Manyara.

**Writing – review & editing:** Tadios Manyanga, Rudo Chingono, Shane Naidoo, Kate Mattick, Grace Pearson, Opeyemi Babatunde, Niri Naidoo, Kate A. Ward, Celia L. Gregson.

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
