## [Decision Letter · Decision Letter 0]

4 Sep 2024

PONE-D-24-08415Types, design, implementation, and evaluation of nutrition interventions in older people in Africa: a scoping review protocolPLOS ONE

Dear Dr. Manyara,

Thank you for submitting your manuscript to PLOS ONE. After careful consideration, we feel that it has merit but does not fully meet PLOS ONE’s publication criteria as it currently stands. Therefore, we invite you to submit a revised version of the manuscript that addresses the points raised during the review process.

We look forward to receiving your revised manuscript.

Kind regards,

Annesha Sil, Ph.D.

Associate Editor

PLOS ONE

Journal Requirements: When submitting your revision, we need you to address these additional requirements. 1. Please ensure that your manuscript meets PLOS ONE's style requirements, including those for file naming. The PLOS ONE style templates can be found at https://journals.plos.org/plosone/s/file?id=wjVg/PLOSOne_formatting_sample_main_body.pdf and https://journals.plos.org/plosone/s/file?id=ba62/PLOSOne_formatting_sample_title_authors_affiliations.pdf 2. Please provide a complete Data Availability Statement in the submission form, ensuring you include all necessary access information or a reason for why you are unable to make your data freely accessible. If your research concerns only data provided within your submission, please write "All data are in the manuscript and/or supporting information files" as your Data Availability Statement. 3. Please review your reference list to ensure that it is complete and correct. If you have cited papers that have been retracted, please include the rationale for doing so in the manuscript text, or remove these references and replace them with relevant current references. Any changes to the reference list should be mentioned in the rebuttal letter that accompanies your revised manuscript. If you need to cite a retracted article, indicate the article’s retracted status in the References list and also include a citation and full reference for the retraction notice.

Reviewers' comments:

Reviewer's Responses to Questions

**Comments to the Author**

1. Does the manuscript provide a valid rationale for the proposed study, with clearly identified and justified research questions?

Reviewer #1: Yes

2. Is the protocol technically sound and planned in a manner that will lead to a meaningful outcome and allow testing the stated hypotheses?

Reviewer #1: Yes

3. Is the methodology feasible and described in sufficient detail to allow the work to be replicable?

Reviewer #1: No

4. Have the authors described where all data underlying the findings will be made available when the study is complete?

Reviewer #1: Yes

5. Is the manuscript presented in an intelligible fashion and written in standard English?

Reviewer #1: Yes

6. Review Comments to the Author

You may also provide optional suggestions and comments to authors that they might find helpful in planning their study.

Reviewer #1: The protocol addresses an important research question for the region- very welldone on this.

I have a few things the authors may wish to think more about.

Review questions

The use of the phrase "recommended" in the question raises some concerns, how does this fit into the inclusion criteria in the end - recommended by who? I donot see this reflected in the PICO

1) Include a justification for including older persons in long term care facilities and provide more descriptive details on what type of facilities are being considered here and the purpose for which older persons are in these facilities.

2) You may wish to specify what kind of qualitative studies are being targeted for inclusion, is it descriptive, process evaluations or??

3) The data analysis/synthesis section is too brief and unclear. Please provide more details on how review questions will be answered at this level.

4) How will the workshop participants be identified and later contacted and by who? The details of this process are lacking.

7. PLOS authors have the option to publish the peer review history of their article (what does this mean?). If published, this will include your full peer review and any attached files.

Reviewer #1: **Yes: **Eve Namisango

---

## [Author Response · Author response to Decision Letter 0]

25 Sep 2024

We have uploaded a response to each of the comments received. We are not able to respond to them here as we have added a table which may not be added to this box.

---

## [Editor Report · Decision Letter 1]

17 Oct 2024

Types, design, implementation, and evaluation of nutrition interventions in older people in Africa: a scoping review protocol

PONE-D-24-08415R1

Dear Dr. Manyara,

We’re pleased to inform you that your manuscript has been judged scientifically suitable for publication and will be formally accepted for publication once it meets all outstanding technical requirements.

Kind regards,

Laura Kelly

Division Editor

PLOS ONE
---

## [Editor Report · Acceptance letter]

29 Oct 2024

PONE-D-24-08415R1 

PLOS ONE

Dear Dr. Manyara, 

I'm pleased to inform you that your manuscript has been deemed suitable for publication in PLOS ONE. Congratulations! Your manuscript is now being handed over to our production team.

Kind regards, 

on behalf of

Dr. Laura Hannah Kelly 

Staff Editor

PLOS ONE